

# Observing short timescale cloud development to constrain aerosol-cloud interactions

Edward Gryspeerdt[1], Franziska Glassmeier[2], Graham Feingold[3], Fabian Hoffmann[4], and Rebecca J. Murray-Watson[1]

[1]Space and Atmospheric Physics Group, Imperial College London, UK
[2]Department Geoscience and Remote Sensing, Delft University of Technology, Delft, The Netherlands
[3]National Oceanic and Atmospheric Administration (NOAA), Chemical Sciences Laboratory, Boulder, Colorado, USA
[4]Ludwig-Maximilians-Universität, München, Germany

**Correspondence:** Edward Gryspeerdt (e.gryspeerdt@imperial.ac.uk)

**Abstract.** The aerosol impact on liquid water path (LWP) is a key uncertainty in the overall climate impact of aerosol. However, despite a significant effort in this area, the size of the effect remains poorly constrained, and even the sign is unclear. Recent studies have shown that the relationship between droplet number concentration ($N_d$) and LWP is an unreliable measure of the impact of $N_d$ variations on LWP due to the difficulty in establishing causality. In this work, we use satellite observations of the short-term development of clouds to examine the role of $N_d$ perturbations in LWP variations.

Similar to previous studies, a increase followed by a general decrease in LWP with increasing $N_d$ is observed, suggesting an overall negative LWP response to $N_d$ and a warming LWP adjustment to aerosol. However, the $N_d$ also responds to the local environment, with aerosol production, entrainment from the free troposphere and wet scavenging all acting to modify the $N_d$. Many of these effects act to further steepen the $N_d$-LWP relationship and obscure the causal $N_d$ impact on LWP.

Using the temporal development of clouds to account for these feedbacks in the $N_d$-LWP system, a weaker negative $N_d$-LWP relationship is observed over most of the globe. This relationship is highly sensitive to the initial cloud state, illuminating the roles of different processes in shaping the $N_d$-LWP relationship. The nature of the current observing system limits this work to a single timeperiod for observations, highlighting the need for more frequent observations of key cloud properties to constrain cloud behaviour at process timescales.

## 1 Introduction

Cloud processes, particularly precipitation and entrainment, depend on the size and number of cloud droplets. Increases in atmospheric aerosols perturb the number concentration of cloud droplets ($N_d$). This increase in $N_d$ can modify the development and properties of a cloud, resulting in "cloud adjustments" to the aerosol perturbation (e.g. Albrecht, 1989). Following increases in anthropogenic aerosol, these cloud adjustments may lead to significant radiative forcings (Forster et al., 2022), although the magnitude (and in some cases the sign) is not currently well constrained (Bellouin et al., 2020).

The impact of aerosol on cloud liquid water path (LWP) is an important component of these adjustments. With a possibility for both increases (Albrecht, 1989) and decreases (Wang et al., 2003; Ackerman et al., 2004) in LWP in response to aerosol,





developing global constraints for the LWP response has proved challenging. High resolution models often produce a decrease in LWP in high aerosol environments through an interaction between aerosol, turbulence and entrainment (Xue and Feingold, 2006; Bretherton et al., 2007) and hence a positive radiative forcing (a warming) that offsets the cooling of the Twomey effect. As cloud adjustments are usually implemented as modifications to precipitation processes, global climate models more often show an increase in LWP (a cooling effect; Malavelle et al., 2017), although this increase is often small (Gryspeerdt et al., 2020).

Due to difficulties using the aerosol optical depth as a proxy for cloud condensation nuclei (CCN; Quaas et al., 2010; Stier, 2016), many recent observational studies have focussed on the $N_d$-LWP relationship as a method for quantifying the aerosol impact on LWP. Although a positive relationship is found in some locations (Han et al., 2002; Murray-Watson and Gryspeerdt, 2022), these studies often identify a negative relationship that would indicate a LWP reduction with increasing aerosol (Michibata et al., 2016; Toll et al., 2019; Gryspeerdt et al., 2019). These studies may be negatively biased (overestimating the warming effect) due to correlated errors in the $N_d$ and LWP retrievals (Gryspeerdt et al., 2019).

In contrast, recent model studies have suggested the $N_d$-LWP relationship derived from exogenous aerosol perturbations (e.g. shiptracks) may be positively biased (underestimating the warming effect), if they don't consider the temporal development of the perturbation (Glassmeier et al., 2021; Gryspeerdt et al., 2021a). This makes it difficult to use current observational studies to provide a tight constraint on the aerosol impact on LWP.

Identifying the aerosol impact on LWP is particularly challenging as the processes involve feedbacks. An increase in LWP may make precipitation more likely, in turn reducing the $N_d$, increasing droplet sizes and further increasing the likelihood of precipitation (e.g. Jing and Suzuki, 2018). This feedback introduces cycles into the causal network, complicating the process of isolating the $N_d$ impact on LWP (McCoy et al., 2020). Temporal information about cloud development provides one way out of this problem (Pearl, 1994; Mülmenstädt and Feingold, 2018), related to the concept of Granger causality (does knowledge of the aerosol environment at time $t_0$ enable you to better predict the cloud state at $t_1 > t_0$?).

The short-term development of clouds has previously been used to investigate aerosol effects (Matsui et al., 2006; Meskhidze et al., 2009; Gryspeerdt et al., 2014). By ensuring that the high and low aerosol populations of clouds have the same initial state, the initial retrieval biases and spurious correlations are reduced, uncovering the impact of aerosol on the cloud development. However, spurious correlations can swamp the aerosol signal if the initial state of the cloud is not controlled for (e.g. Gryspeerdt et al., 2014).

Glassmeier et al. (2021) demonstrates a different pathway for the use of temporal information. With multiple model simulations following the evolution of different initial cloud states, they produce a "flowfield", allowing nocturnal cloud evolution to be traced forward, beyond the length of any individual simulation. In this work, we apply a similar technique to satellite observations, using the development of clouds over short timescales (<6 hours) to examine the role of $N_d$ in controlling the LWP. We place a particular emphasis on the controlling the initial state of the cloud to account for the impact of existing co-variations on the development of the $N_d$-LWP relationship. These results demonstrate that LWP evolves differently depending on the initial $N_d$ perturbation and that instantaneous measurements of the $N_d$-LWP relationship may not accurately capture the $N_d$ impact on LWP in liquid clouds.



## 2  Methods

The $N_d$ and LWP data in this work are primarily from the two MODIS instruments onboard the Aqua and Terra satellites for
ten years (2011-2020 inclusive). The level 2 (1km resolution) collection 6.1 cloud product (MOD06_L2; Platnick et al., 2017)
is used to calculate the $N_d$, following the sampling criteria outlined in Grosvenor et al. (2018) and (Gryspeerdt et al., 2021b,
sampling strategy G18). The $N_d$ is calculated assuming a adiabatic cloud (Quaas et al., 2006) for these selected pixels. The LWP
is calculated using all the available liquid pixels, as restricting the LWP retrieval to only the pixels used for the $N_d$ calculation
biases it towards higher optical depths, leading to a high LWP bias against passive microwave LWP (Gryspeerdt et al., 2019).
This data is aggregated to a 1° by 1° grid separately for each MODIS instrument and each day. Note that aggregation is
performed using the collection 6 "definition of a day" to ensure that the relative temporal ordering of the data is preserved near
the dateline as closely as possible (Hubanks et al., 2020).

Each satellite and day is treated separately. The two daytime MODIS overpasses (at approximately 10:30 LST for Terra and
13:30 LST for Aqua) provide the necessary temporal development (over an approximately three hour period)to estimate the
impact of $N_d$ on LWP using a difference-in-differences method (Fig. 1).

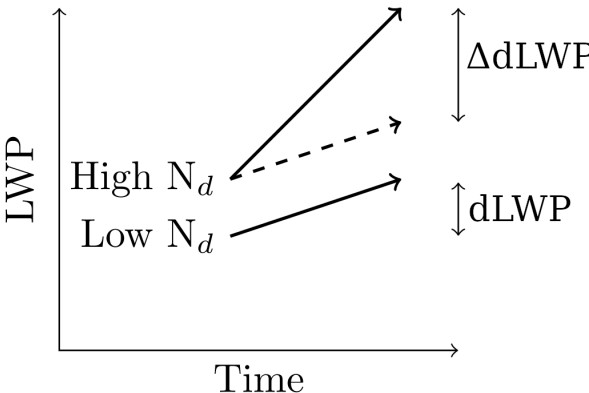

**Figure 1.** A schematic of the difference-in-differences method, showing how the $N_d$ impact on LWP development is identified using the
temporal development of the cloud field. This example shows a positive ΔdLWP.

The difference in properties between the Aqua and Terra overpasses is indicated with a "d", i.e. dLWP is the afternoon
(Aqua) LWP minus the morning (Terra) LWP. Separating the population of clouds into two groups (those with a starting $N_d$
more and less than the median value of $60 \, \mathrm{cm^{-3}}$), $\Delta_{N_d}$dLWP is defined as the difference in the dLWP for the above and below
$60 \, \mathrm{cm^{-3}}$ $N_d$ groups. As the evolution in this work is always separated by the other variable, the subscript on the $\Delta$ is omitted.
A positive ΔdLWP means that the high $N_d$ population gained more (or lost less) LWP over the 3 hour period than the low $N_d$
population. This would suggest a positive $N_d$ impact on LWP (and hence a negative radiative effect for LWP adjustments). The
$N_d$-LWP relationship is also characterised using the "sensitivity", the slope of the linear regression between the two values in
log-log space ($\frac{d \ln LWP}{d \ln N_d}$ Feingold, 2003). Similar difference in differences calculations are performed for the $N_d$ evolution, using
populations above and below $60 \, \mathrm{g \, m^{-2}}$ ($\Delta dN_d$).





To account for motion in the cloud field, the field is advected using ERA5 reanalysis fields at 1000 hPa, with this level
selected as it can accurately predict the locations of shiptracks given the location of individual ships (Gryspeerdt et al., 2021a).
The expected motion over three hours is often less than 1°, so the advection step is calculated at a 0.25° by 0.25° resolution.
Each of these quarter-degree gridboxes is advected following the ERA5 winds. The end locations of these trajectories are used
to sample the Aqua data and this re-sampled data is aggregated to 1° by 1° resolution for this analysis. Pixels with no cloud
retrieval (either morning or afternoon) are removed from this analysis. This means that the results in this work consider only
the development of in-cloud LWP, matching the decomposition of the forcing into $N_d$ (Twomey), LWP and CF components in
previous work (Bellouin et al., 2020).

   The CCCM (CERES-CloudSat-CALIPSO-MODIS) combined product (Kato et al., 2010) is used to examine the role of
precipitation on the LWP and $N_d$ development. The CloudSat precipitation flag (Haynes et al., 2009) from CCCM is used to
calculate the probability of precipitation as a function of LWP and $N_d$. Only oceanic, liquid phase data are used over the period
2007-2011 (inclusive). Both liquid precipitation and drizzle are considered as precipitating for the purposes of this analysis.

## 3   Results

### 3.1   LWP development

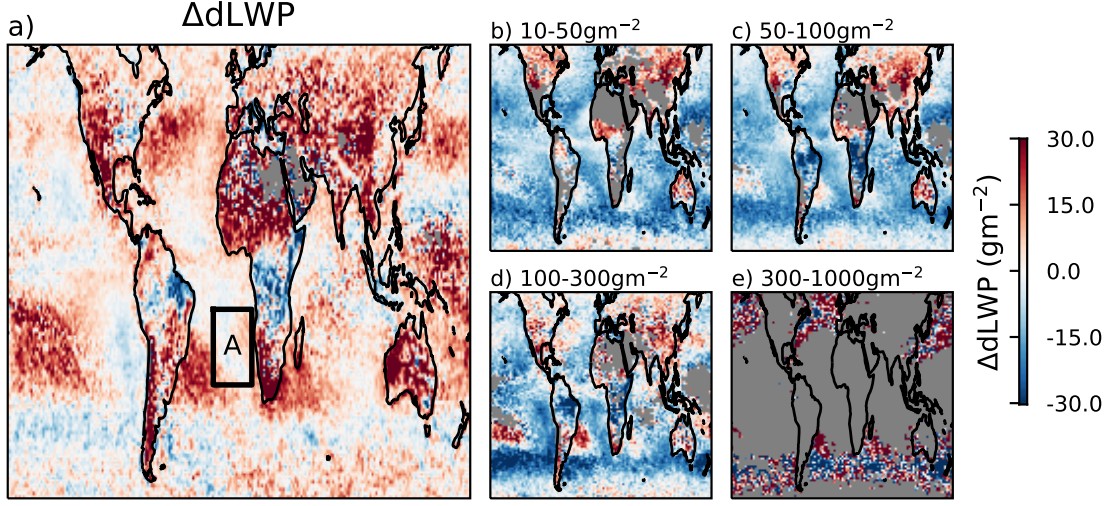

**Figure 2.** The difference in dLWP between high and low initial $N_d$ populations (ΔdLWP). Red indicates a more positive dLWP for the high
$N_d$ population ($N_d$ >60 cm$^{-3}$). (a) Includes all available data, b-e) Only considering cases where the initial LWP is within the specified
bounds.

   In many regions, ΔdLWP is positive (Fig. 2a), particularly away from the stratocumulus decks This suggests an increase (or
weaker decrease) in LWP at higher $N_d$. The ΔdLWP is larger at mid-latitudes and towards the west of the subtropical oceanic





regions (where the environment is typically more unstable). This result, with higher LWPs in higher $N_d$ cases, is in contrast to previous studies looking at large-scale statistics which typically show a reduction in LWP as $N_d$ increases (e.g. Michibata et al., 2016; Gryspeerdt et al., 2019; Possner et al., 2020).

In this case, the positive ΔdLWP is an artifact of the strong initial negative $N_d$-LWP relationship (Fig. 4a). Over the three
100    hour observation period, cases with a low initial LWP will tend to increase in LWP, whilst those with a high initial LWP will decrease (a concept known as regression to the mean), returning towards an LWP steady state (Hoffmann et al., 2020). Due to the negative $N_d$-LWP relationship, cases with a high initial $N_d$ have a corresponding low initial LWP and so might be expected to have a more positive dLWP than the low $N_d$ population, producing the apparent positive $N_d$ impact on LWP in Fig. 2a. This would happen even without a causal $N_d$ impact on LWP. By binning by the initial cloud state, this ensures that the high and
low $N_d$ populations start with the same LWP.

Controlling for the initial LWP uncovers a negative ΔdLWP in most regions and under most initial conditions, suggesting that an increase in $N_d$ leads to a lower LWP over time (Fig. 2b-e). A positive ΔdLWP remains over land, particularly in cases with a low starting LWP. This might be related to convective invigoration (e.g. Koren et al., 2014), but the $N_d$ retrieval is less accurate over land due to the lower adiabaticity of convective clouds (Gryspeerdt et al., 2021b), leading to a low confidence in
this result. The different apparent $N_d$ impact on LWP highlights the importance of controlling for the initial cloud state when looking at cloud development.

### 3.2   $N_d$ **development**

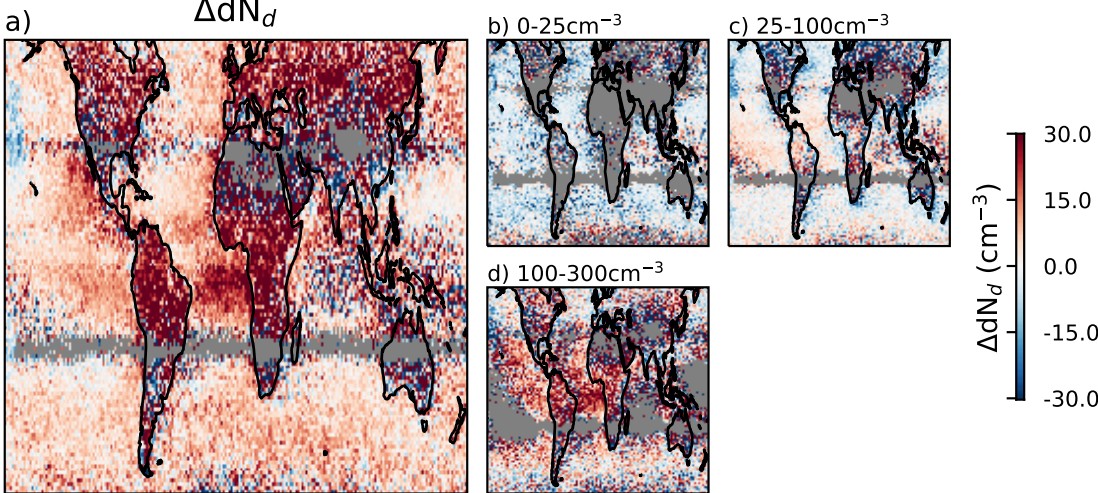

**Figure 3.** As Fig. 2, but showing the difference in $N_d$ evolution between clouds with an initial LWP higher and lower than $60 \mathrm{gm}^{-2}$. (a) shows all data, while (b-d) show cases where the initial $N_d$ is restricted to the specified range. Note the larger fraction of missing data due to the more stringent sampling constraints on the $N_d$ retrieval.



Cloud and aerosol processes also modify the $N_d$ over the three hour period between the Terra and Aqua overpasses. Following Wood (2012), three processes are expected to dominate changes in $N_d$ away from strong aerosol sources: CCN entrainment or

115 dilution from mixing with the free troposphere, CCN production through sea salt emission, and the depletion of CCN through wet scavenging. Of these, wet scavenging is expected to have the strongest link to LWP, as precipitation is more common at high LWP (e.g. L'Ecuyer et al., 2009; Sorooshian et al., 2009) and so produce a negative $\Delta dN_d$. A positive $\Delta dN_d$ is instead found across most of the globe (Fig. 3a).

This positive $\Delta dN_d$ is strongest over land and in regions downwind of continents (e.g. the south east Atlantic, Sea of Japan

and Tasman Sea; Fig. 3a) is primarily driven by the $\Delta dN_d$ in polluted conditions (Fig. 3d).

In clean conditions ($N_d > 25\,\mathrm{cm}^{-3}$), a larger LWP results in a more negative $\Delta N_d$. This is as expected from wet scavenging, where an increased LWP increases the probability of precipitation (e.g. Ludlam, 1951; Sorooshian et al., 2009; L'Ecuyer et al., 2009), reducing the $N_d$. In these cases, increasing the LWP increases the probability of precipitation, decreasing the $N_d$ more strongly over time for higher initial LWP. This negative $\Delta dN_d$ is also visible near coastlines, particularly in the northern

hemisphere, for moderately polluted cases (Fig. 3c). The regions of negative $\Delta dN_d$ in this case are typically in the more polluted locations. Positive $\Delta dN_d$ values are seen in the tropics.

In many polluted regions, particularly off the West coast of Africa, there are strong positive $\Delta dN_d$ values, which drive the overall $\Delta N_d$ response to LWP. This is opposite to the impact expected from wet scavenging, as precipitation becomes relatively rare for $N_d$ values above $100\,\mathrm{cm}^{-3}$, except at the largest LWPs (e.g. Fig. 5a-c). For the majority of both the high and the low

LWP populations, the probability of precipitation is close to zero, obscuring the role of wet scavenging.

With precipitation uncommon, $\Delta dN_d$ at high $N_d$ isolates the impacts of CCN entrainment and production in driving $\Delta dN_d$. With free troposphere CCN being a major CCN source (Wood et al., 2012), this increase in $N_d$ at high LWP is likely due to the warm, moist air that often accompanies biomass burning aerosol (Adebiyi et al., 2015). When above the cloud, the moist smoke layer reduces cloud top cooling, limiting the LWP and providing no extra $N_d$. However, when the moist smoke layer

is in contact with the cloud, LWP increases and the additional source of CCN gradually increases the $N_d$ (Yamaguchi et al., 2015). This source effect is only visible in non-precipitating situations, as precipitation typically dominates the $N_d$ budget in marine locations (Wood et al., 2012).

### 3.3 $N_d$-LWP development

The maps in Figs. 2 and 3 show a global variation in cloud development as a function of initial $N_d$ and LWP, but are a relatively

coarse tool, hiding much of the complexity of the $N_d$-LWP temporal development. Fig. 4 shows how the LWP and $N_d$ change over three hours (dLWP, $dN_d$) as a function of the initial LWP and $N_d$ for a region within the south east Atlantic stratocumulus deck (region A in Fig. 2a).

This region displays the "inverted-V" pattern for LWP as a function of initial $N_d$ (Fig. 4a), with a positive slope at low $N_d$ (consistent with precipitation suppression) and a negative slope at high $N_d$, consistent with increased entrainment at high $N_d$

(Gryspeerdt et al., 2019). In contrast to this inverted-V, the $N_d$ normalised by LWP (Fig. 4b) shows a monotonic decrease in $N_d$ as the LWP increases, becoming constant at high LWP.



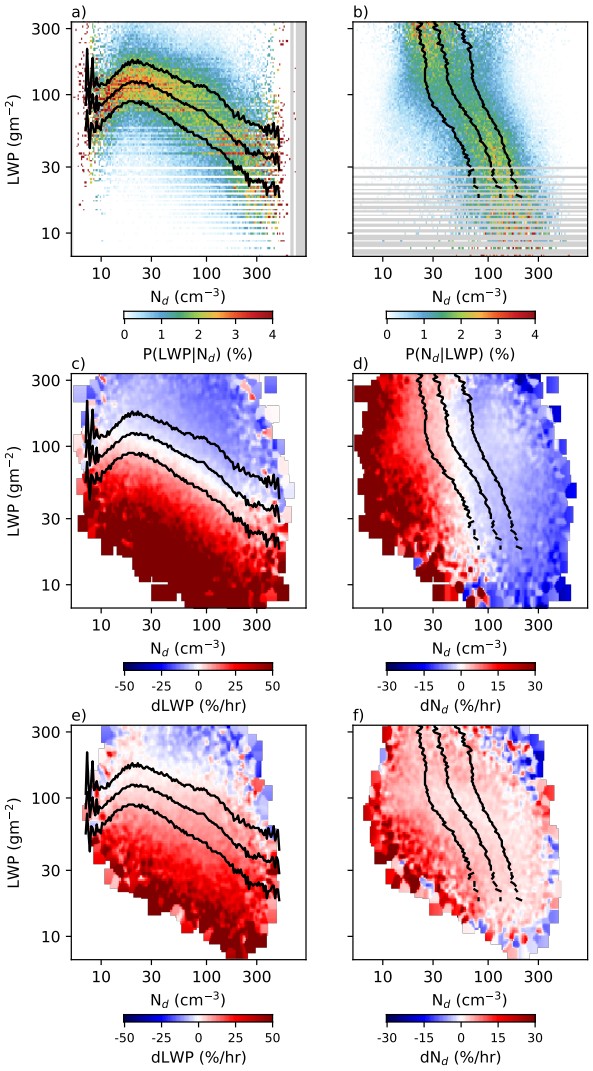

**Figure 4.** $N_d$-LWP development fields for Namibian stratocumulus region (A in Fig. 2a). a) the probability of observing an initial LWP for a given initial $N_d$ (P(LWP|$N_d$)), b) P($N_d$ |LWP). c,d) Red is a positive (c) dLWP or (d) d$N_d$ for a given initial (morning/Terra) $N_d$ and LWP and blue is a negative (decrease in LWP or $N_d$). The fields are smoothed with a Gaussian window filter. e,f) as (c,d) but binned using the final (afternoon/Aqua) $N_d$ and LWP. The black lines are at 25, 50 and 75th percentiles.

The LWP evolution (Fig. 4c) reflects the initial LWP distribution in (Fig. 4a). For a given $N_d$, positive dLWP is found at lower initial LWPs and a negative dLWP at higher initial LWP. This relationship is also a clear function of the initial $N_d$, with the dLWP=0 contour reducing as the initial $N_d$ increases. The positive dLWP values at lower LWP are much stronger than the negative values found at high LWP. This overall pattern is very similar to that obtained from LES modelling (Hoffmann et al.,



2020). The strong dependence on dLWP on LWP highlights the importance of considering the initial LWP when investigating temporal LWP changes.

Similarly, the $N_d$ evolution (Fig. 4d) reflects the $N_d$ distribution in (Fig. 4b). Positive $dN_d$ values are found at low initial $N_d$ and negative values at high $N_d$. This means that over time, the $N_d$ would be expected to collapse to the $dN_d = 0$ contour. The $dN_d = 0$ contour is not in the same location as the peak of the $N_d$ distribution in Fig. 4b, indicating that the $N_d$ is not in equilibrium.

The dLWP field (Fig. 4c) is similar to that found in model studies (Glassmeier et al., 2021), collapsing down to an approximate inverted-V shape, while the $dN_d$ behaviour is quite different. In Glassmeier et al. (2021), $dN_d$ is negative at low $N_d$ (due to wet scavenging depleting $N_d$ over time) and positive at high $N_d$ (due to an aerosol source), causing the $N_d$ to diverge over time. This is in contrast to the $dN_d$ behaviour in Fig. 4d, where the $N_d$ flow pattern converges towards an equilibrium value. Some of these differences may be explained by differences in the time of day (Glassmeier et al., 2021,  was nocturnal whilst these are daytime satellite retrievals), but meteorological controls on $N_d$ may also play a role.

## 3.4 Meteorological controls on $N_d$ development

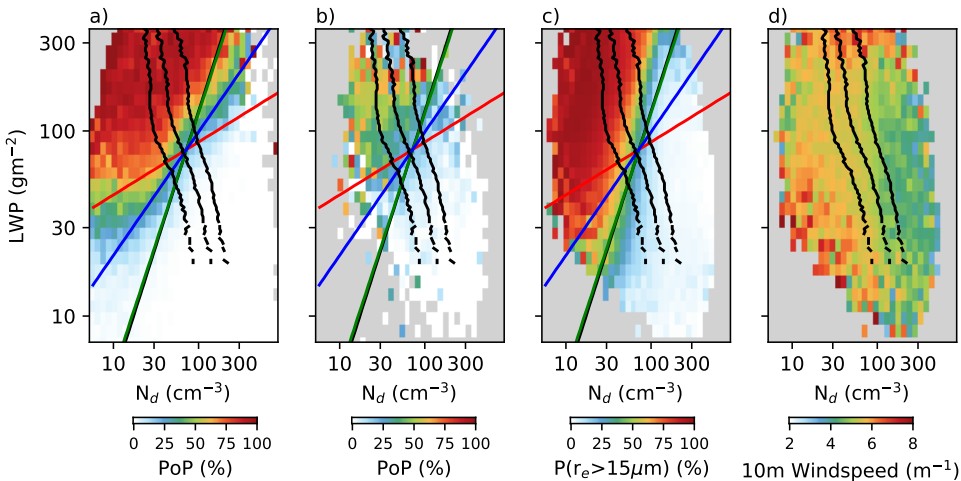

**Figure 5.** a) The CloudSat probability of precipitation as a function of $N_d$ and LWP at a pixel (1km) resolution. The lines are contours of constant autoconversion rate from Tripoli and Cotton (1980, red), Liu and Daum (2004, blue) and Khairoutdinov and Kogan (2000, green) . b) as (a), but for $N_d$ and LWP at a $1°$ by $1°$ resolution. c) the proportion of liquid $r_e$ retrievals >15μm for $N_d$ and LWP at a $1°$ by $1°$ resolution. d) the average ERA5 windspeed as a function of LWP and $N_d$.

The primary controls on $N_d$ development are: the production of CCN through sea-salt emission (increasing $N_d$), wet scavenging (reducing $N_d$) and the mixing with free-troposphere air (increasing or decreasing $N_d$; Wood, 2012). While modelling wet scavenging, Glassmeier et al. (2021) include a constant aerosol source, which does not depend on these environmental controls in the same way.



Sea salt production depends strongly on the local windspeed and is correlated to the $N_d$. This relationship is not linear. While windspeed (or sea-salt production) are positively correlated with $N_d$ at low windspeeds, decreases in $N_d$ have been observed at high windspeeds and sea-salt burdens, potentially due to the impact of giant CCN (Gryspeerdt et al., 2016; McCoy et al., 2018). This is reflected in Fig. 5d, where the highest windspeeds (and so positive impact on $dN_d$) are found at low $N_d$ values, with little dependence on the LWP. Sea salt production therefore contributes to the positive $dN_d$, primarily at low $N_d$ values, where the windspeed is strongest.

Free troposphere mixing exchanges aerosol with a large reservoir, which has the effect of bringing the $N_d$ back towards the free troposphere value. At low $N_d$ values, this produces a positive $dN_d$. At high $N_d$ values, this produces a negative $dN_d$. Some correlation to LWP is possible, as the free-troposphere CCN can be correlated to the humidity, which is itself correlated to the LWP for underlying marine stratocumulus (e.g. Fig. 3d, Gryspeerdt et al., 2019). However, the $dN_d$ produced by free-tropospheric mixing is largely independent of LWP (Fig. 4d).

The impact of wet scavenging is observed in the upper-left quadrant of Fig. 4d, but it does not dominate the $dN_d$. Following the $dN_d =0$ contour, at higher LWP values, this contour shifts to lower $N_d$. However, wet scavenging is not strong enough to produce a negative $dN_d$ across the precipitating region of Fig. 4d. This is due to the sub-grid variability in cloud properties at $1°$ by $1°$ resolutions.

When calculated at a pixel level, the probability of precipitation (PoP) is a strong function of both the LWP and $N_d$ (Fig. 5a), with low $N_d$, high LWP cases having a PoP of over 80%. At this resolution, the transition from precipitating to non-precipitating is sharp and close to linear in log-log space. Assuming an adiabatic liquid water content profile, the PoP edge is parallel to contours of the autoconversion rate from the Liu and Daum (2004) scheme. At a 1 km resolution, precipitation becomes rare as LWP drops below $30\,\mathrm{g\,m^{-2}}$. The mean state $N_d$ becomes relatively insensitive to LWP increases above this boundary (Fig. 4b; Fig. 5a, black lines), but a clear transition such as this might be expected produce a clear boundary in Fig. 4d along this edge. Although precipitation is important for $N_d$ evolution, no clear transition in the $N_d$ evolution is observed.

Precipitation is a non-linear function of $N_d$ and LWP; sub-grid variability in cloud water and $N_d$ modifies the autoconversion rate (Zhang et al., 2019). This is clear when calculating the PoP at $1°$ by $1°$ resolution (Fig. 5b). While the high $N_d$, low LWP cases are still primarily non-precipitating, the probability of precipitation for the low $N_d$ cases peaks at around 50% (so only half of CloudSat rays in liquid cloud conditions are precipitating). Precipitation is even observed in cases with high $N_d$.

Similar variability is observed when using the cloud top effective radius ($r_e$) as a measure of precipitation (Fig. 5c). Using the probability of a 1 km pixel having an $r_e >15\mu$m, a much stronger relationship between $N_d$ and precipitation is observed at $1°$ than at 1km, with this transition being parallel to the Khairoutdinov and Kogan (2000) autoconversion rate. This transition is also less complete, with the probability of finding an $r_e >15\,\mu$m not falling below a few %. This sub-grid variability decreases the precipitation contrast between the precipitating and non-precipitating regions, obscuring the impact of wet scavenging in these results.

The lack of a clear dividing line into precipitating and non-precipitating regions in the $N_d$-LWP plot blurs the impact of wet scavenging, in contrast to high resolution model simulations (Hoffmann et al., 2020). When combined with an aerosol source that is weakly dependent on $N_d$ (the sea-salt source) and free-troposphere entrainment that can act as an aerosol sink in high





$N_d$ cases, this produces a $N_d$ state that converges towards a stable state (Fig. 4d), rather than the diverging, unstable state seen in model studies (Glassmeier et al., 2021). This blurring effect also hides the second potential stable state at high $N_d$ seen in

previous model studies (Baker and Charlson, 1990). Although wet scavenging has a relatively subtle effect on the $N_d$ flowfield in Fig. 4d, it still has a clear effect on the equilibrium $N_d$. As the LWP increases into the precipitating regime, the d$N_d$ =0 contour shifts from around 60 to closer to 30 cm$^{-3}$. This is consistent with the results of Wood (2012), who demonstrated the key role of wet scavenging in setting the mean $N_d$.

### 3.5 Implications for the LWP response to $N_d$

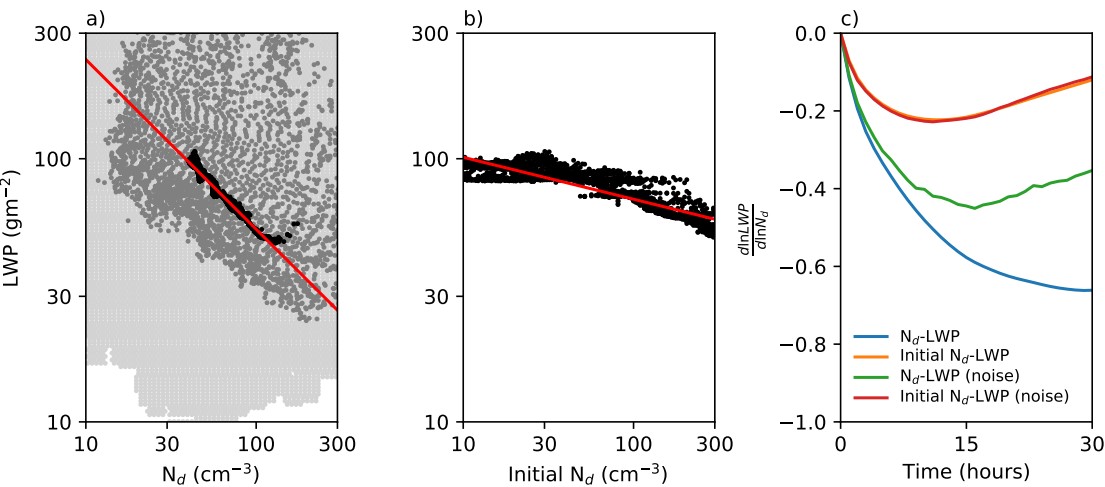

**Figure 6.** Temporal evolution of sensitivity assuming a constant flow field for region A in Fig. 2a. (a) The instantaneous $N_d$-LWP relationship after 3 hours (grey) and 24 hours (black), from a distribution of points that have no initial $N_d$-LWP relationship (light grey). (b) the relationship between the initial $N_d$ and LWP for the same data. (c) $\frac{d\ln LWP}{d\ln N_d}$ as a function of time measured using the instantaneous relationship (blue) and initial $N_d$ (orange). Green and red are these relationships with Gaussian noise applied to the flowfield evolution. The red and orange lines overlap.

Combining the two development fields in Fig. 4c,d specifies the function $\Delta LWP, \Delta N_d = f(LWP, N_d)$. This function is used to evolve a joint ($N_d$, LWP) distribution, producing an $N_d$-LWP slope and allowing these flowfields to be compared to previous studies that identified $\frac{d\ln LWP}{d\ln N_d}$ (Han et al., 2002; Michibata et al., 2016; Gryspeerdt et al., 2019). Although this makes the somewhat unrealistic assumption that the function remains constant with time, it allows for a comparison with previous studies of the instantaneous $N_d$-LWP relationship and provides a method to examine the impact of feedbacks in the system.

With an initial array of (LWP, $N_d$) points that are sampled so that there is no initial $N_d$-LWP relationship, these points are then stepped forward using the fields shown in Fig. 4. After eight steps (approximately 24 hours), this produces the strong negative $N_d$-LWP relationship shown in Fig. 6a. The temporal development of the $N_d$-LWP sensitivity for this population is shown in Fig. 6c by the blue line, with the sensitivity reaching a minimum of -0.7 at around 15 hours (5 timesteps). Similar to





recent model studies (Glassmeier et al., 2021), this sensitivity is noticeably stronger than previous observational studies, which
typically are smaller than -0.4 and closer to -0.1 (Toll et al., 2019; Gryspeerdt et al., 2019).

One complicating factor in measuring the $N_d$ impact on LWP is that the $N_d$ also evolves with time. This means that the
measured sensitivity at each timestep is the combination of $N_d$ impacts on LWP along with feedback processes that modify
the $N_d$. As the $N_d$ evolution also acts to create a negative $N_d$-LWP relationship, the instantaneous $N_d$-LWP relationship at a
given timestep is not a good measure of the causal $N_d$-LWP relationship. To minimise this issue, we also show the relationship
between the *initial $N_d$* and the evolved LWP after 24 hours (Fig. 6b). This shows a weaker sensitivity, with a minimum around
10-12 hours before decreasing (absolute value) again with time (Fig. 6c, orange line). At -0.2 the peak sensitivity is less negative
than the instantaneous $N_d$-LWP sensitivity (blue line), but still more negative than that obtained in many observational studies.
This temporal development is due to the weaker LWP reduction in low $N_d$ cases (Fig. 4c). An approximately linear relationship
is formed initially, before additional LWP reductions generate the weak $N_d$-LWP relationship below 30cm$^{-3}$, weakening the
overall relationship (Fig. 6b).

A further complication comes from the impact of other, uncorrelated factors. The $N_d$ and LWP alone are unlikely to be the
only factors governing the evolution of $N_d$ and LWP in the cloud field. If these other factors are uncorrelated to the current $N_d$
and LWP, they can be represented as noise on the dLWP and d$N_d$ terms. Adding noise to the (LWP,$N_d$) evolution weakens the
instantaneous sensitivity (Fig. 6c, green line), primarily by widening the $N_d$ range to include the non-linear regions at low and
high $N_d$. The magnitude of the weakening effect depends strongly on the strength of the noise. As the measurements used in
this work already contain a noise component, this might suggest that the sensitivities obtained in this work are too weak - a
more accurate measure of the temporal evolution of clouds might produce stronger sensitivities.

## 3.6 The global distribution

The dLWP and d$N_d$ fields in Fig. 4 and their evolution (as in Fig. 6) vary across the globe, becoming positive in some regions.
Fig. 7a shows the average sensitivity for the period 18-24 hours after the integration is started. The sensitivity is strongly
negative over almost all ocean regions (consistent with Fig. 2), while a slight positive sensitivity is observed over land. The
stratocumulus sensitivity is above -0.6 in many regions, which would lead to a complete cancellation of the forcing from the
Twomey effect and a positive effective radiative forcing from aerosol-cloud interactions in these locations (Glassmeier et al.,
2021).

As noted in the previous section, the instantaneous sensitivity incorporates feedbacks on the $N_d$ (such as wet scavenging)
that act to steepen the $N_d$-LWP relationship. Fig. 7b shows the $N_d$-LWP sensitivity calculated using the initial $N_d$, which is
closer to the causal impact of $N_d$ on LWP. There are many similarities between the spatial distributions, with more strongly
negative sensitivities in the stratocumulus decks and positive sensitivities over land. These sensitivities support some previous
conclusions, with negative sensitivities in stratocumulus regions (Toll et al., 2019) and a weak negative sensitivity downwind
of Hawaii (Gryspeerdt et al., 2019). The overall magnitude of the sensitivity is much weaker, peaking close to -0.2 in the
stratocumulus decks. This $N_d$-LWP sensitivity would offset around half of the Twomey effect, with a reduction in LWP and

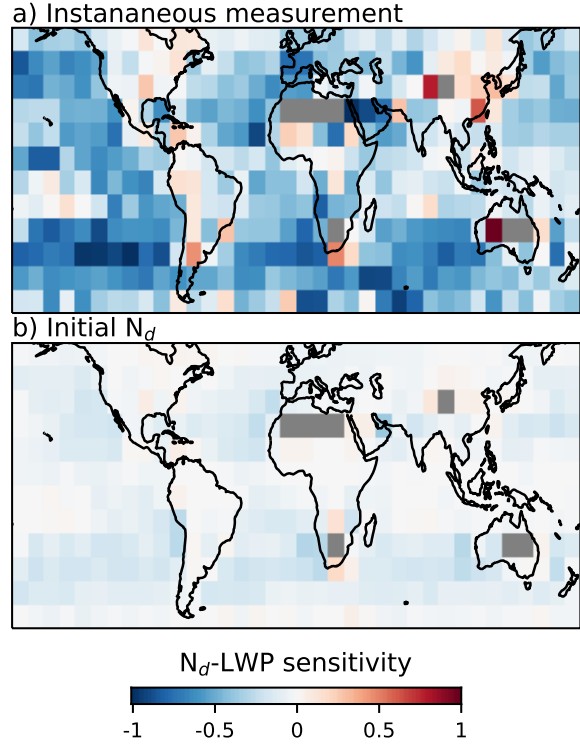

**Figure 7.** a) The instantaneous sensitivity averaged over the period 18-24 hours for each 10° by 10° degree, b) the equivalent using the *initial* $N_d$ for calculating the sensitivity.

a positive radiative forcing stronger than that derived from shiptracks (Toll et al., 2019), but weaker than that derived from MODIS data alone (Gryspeerdt et al., 2019).

## 4   Discussion

While these results show that the short-term behaviour of the LWP and $N_d$ is consistent with the impacts of wet scavenging, CCN production and free-troposphere mixing, some uncertainties in this work remain, particularly around the impact of retrieval uncertainties, the specification of the initial state for integration and the impact of factors that remain unaccounted for.

Systematic biases in retrievals have long been an issue with observation-based aerosol-cloud studies (e.g. Quaas et al., 2010). Studying the temporal development of a scene can reduce these biases, as they are the same for both the initial and final state

and so do not impact dLWP or d$N_d$ (Fig. 1). However, temporal development is subject to a different class of biases created by random retrieval errors, namely regression to the mean. If a random error results in a high bias to the LWP, the later second retrieval is very likely to be smaller. This creates a negative dLWP at high LWP and a positive one at low LWP (and similar for $N_d$), biases which are not removed even by averaging over large datasets.





However, if regression to the mean were driving the results in this work (such as the flowfields in Fig. 4), the flowfields would
look the same if they were calculated in either direction - that is binning dLWP and d$N_d$ by the final LWP and $N_d$. Figs. 4e,f
show the results of this backward flowfield, calculating dLWP and d$N_d$ relative to the *final* LWP and $N_d$. The result is different
to the forward flowfields in Figs. 4c,d. The dLWP=0 line is at a much higher LWP for the backward flowfield when compared
to the forward flowfield, with only a few negative values observed at very high LWPs. The difference in the d$N_d$ field is even
clearer, almost no negative d$N_d$ values are observed. While this does not completely rule out the impact of retrieval biases and
the regression to the mean effect, it builds confidence that these results are not just a statistical artifact caused by random biases
in the LWP and $N_d$ retrievals.

It is also not yet clear how best to use these flowfields to calculate a final sensitivity value. In this work, we assume that
the flowfield is evenly populated and integrate until a slope in the data becomes clear. Should all initial points in the flowfield
be given equal weighting? How can this best be compared to more traditional calculations of the sensitivity? The inclusion
of noise into the integrations also reduces the sensitivity obtained. What is the appropriate level of noise to include? Do the
flowfields remain constant long enough for this technique to be valid? The analysis of short-term cloud development along
trajectories using geostationary data provides one pathway to answering these questions and will be explored in future work.

## 5 Conclusions

The impact of correlated errors in $N_d$ and LWP retrievals makes observed $N_d$-LWP relationships difficult to interpret. The
response of LWP to aerosol perturbations (such as from ships or industry) provides one potential solution to this, but only if
time development is taken into account.

In this work, we look at the short-term development of LWP and $N_d$ as a function of the initial state, between overpasses
of MODIS instruments (approximately 3 hours). Controlling for the initial state reduces the impact of these correlated errors,
showing that the LWP and $N_d$ evolution is highly dependent on the initial state. The instantaneous $N_d$-LWP correlation is
strong enough to generate spurious relationships between the $N_d$ and LWP development if it is not accounted for (Fig. 2a), but
once it is, there is clear evidence of a decrease in LWP at higher $N_d$ levels (Fig. 2). A wet-scavenging effect, reducing $N_d$ as
LWP increases, is only visible under low $N_d$ conditions. In high $N_d$ environments, $N_d$ tends to increase as the LWP increases,
potentially due to a meteorological covariation between CCN sources and airmass properties (Fig. 3).

Binning these short term changes in LWP and $N_d$ as a function of the initial LWP and $N_d$ can represent the cloud development
in more detail (Fig. 4). Although these fields are unlikely constant in time, integrating them forward can convert these three-
hourly development values into a sensitivity suitable for comparing to previous work (Fig. 6). This produces a strongly negative
$N_d$-LWP relationship similar to model studies (Glassmeier et al., 2021), although the evolution of the $N_d$ complicates the
interpretation of the $N_d$-LWP relationship. Using the initial $N_d$ to calculate the $N_d$-LWP sensitivity accounts for these feedback
processes, resulting in a weaker sensitivity of LWP to $N_d$ variations. This sensitivity varies globally; although it is stronger in
stratocumulus regions, it is still weaker than the sensitivity calculated using instantaneous MODIS data.





While this work demonstrates a potential method for accounting for feedbacks when evaluating the $N_d$-LWP relationship, it is still affected by potential retrieval biases in the $N_d$ and LWP retrievals that could affect the quantification of the initial state. To accurately quantify the aerosol impact on LWP, variability in the $N_d$-LWP relationship would have to be accounted for. The diurnal cycle and local meteorological conditions have an impact on LWP evolution and $N_d$, likely affecting the results in Figs. 6c and 7). Geostationary satellites provide a natural pathway forward, although night-time retrievals of cloud properties are challenging. Future work should also account for the possibility these relationships are not constant under warming (Zhang et al., 2022; Murray-Watson and Gryspeerdt, 2022).

Although the magnitude of the $N_d$-LWP relationship derived here is only indicative of the $N_d$ impact on LWP, this work provides a pathway to make use of geostationary observations for constraining cloud processes. It also highlights that the instantaneous $N_d$-LWP relationship measured along a trajectory may not be a good measure of how the LWP is responding to $N_d$ variations. It is vital that trajectory and temporal evolution based studies have the same initial conditions if they are to successfully isolate the aerosol impact on cloud properties and development.

*Data availability.* The MODIS data was obtained through the Level 1 and Atmosphere Archive and Distribution System (LAADS). The ERA5 data was obtained through the Copernicus Climate Data Store.

*Competing interests.* FG and GF are Associate Editors of ACP. The authors declare they have no other competing interests.

*Acknowledgements.* EG was supported by a Royal Society University Research Fellowship (grant no. URF/R1/191602). FH acknowledges support from the Emmy Noether program of the German Research Foundation (DFG) under grant HO 6588/1-1. FG acknowledges support from The Branco Weiss Fellowship – Society in Science, administered by ETH Zuürich, and from a Veni grant of the Dutch Research Council (NWO).



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
