# Peer review of "Observing short timescale cloud development to constrain aerosol-cloud interactions"

_Atmospheric Chemistry and Physics, 2022_

## Author Comment (AC1)

**Reviewer 1**

*: This is a well and clearly written submission that uses Terra and Aqua MODIS observations to analyze aerosol-cloud interactions represented by how liquid water path changes, conditioned by number concentrations. Additionally, they incorporate other combined satellite observations to investigate the role of precipitation in the evolution of liquid water path. They find that liquid water path tends to decrease with increasing number concentration, suggesting the liquid water path adjustment has a warming effect due to aerosols. I especially appreciated the discussion of the overall limitations of using these types of observations and potential paths foreword (i.e. Geostationary satellites). Overall, I didn't find many issues with the manuscript, with most of my questions focusing on different aspects of the methods. Once these are addressed, I expect that this manuscript will be ready for submission.*

**Reply**: We thank the reviewer for their comments, which we address in turn below. Line numbers refer to the diff-ed version.

**Major Comments**

**Methods**: *Why do you use different time periods for the MODIS (2011-2020) and the CCCM combined product (2007-2011) that do not overlap?*

**Reply**: The range has now been extended to 2007-2020. This has made some small changes to some of the plots (including filling in some missing regions), but does not impact the conclusions.

**Lines 80-82**: *Your analysis does not involve ship tracks, so do you only mention ship tracks here because to justify your tracking method?*

**Reply**: Yes, the mention of shiptracks here is due to the use of the 1000hPa wind data to perform the advection step. Winds at this altitude match shiptrack location well over 10 hours or more, so are suitable for use for advecting the clouds over the relatively short 3 hour period between the Terra and Aqua overpasses. The advection has a small impact on the results, but is included for completeness. This sentence has been modified to read "at 1000hPa. This level is selected as it is able to accurately predict the locations of shiptracks given the individual ships, confirming it is suitable for calculating cloud advection over short timescales (Gryspeerdt et al, 2021)."

**Lines 88 - 91**: *CloudSat is limited to a much narrower swath width than MODIS, so how representative is the CloudSat precipitation flag in a 1x1 box?*

**Reply**: Cloudsat indeed represents a smaller fraction of the gridbox than MODIS does, which can lead to representation issues and a single CloudSat measurement is clearly not representative of the precipitation frequency across a 1x1 degree box. However, we have included 4 years of data in Fig. 5, which ensures that each cloud condition ($N_d$, LWP) has a sufficient number of retrievals. As noted by previous work (Stephens et al., 2010), these probabilities of precipitation are not suitable for comparing directly to models due to spatial averaging

issues. We cannot apply the averaging method form Stephens et al. (2010) as we calculate PoP at a cloudsat ray scale, rather than a gridbox scale. The difference in the calculation (and a cautionary note about comparisons with models) is now included at L95

**Minor Comments**

**Line 6**: *"a increase" should be "an increase"*
**Reply**: That is a bit embarrassing, thanks for spotting it. Amended

**Line 99**: *Figure 4 is referenced before Figure 3.*
**Reply**: The referencing has been modified to correct this. To keep the figures close to where they are primarily referenced in the text, this figure reference has been replaced with references to previous studies that identify a link between $N_d$ and LWP (Han et al., 2002; Michibata et al., 2016; Gryspeerdt et al., 2019).

**Line 121**: *Should "clean conditions ( Nd > 25 cm-3)" be clean conditions ( Nd < 25 cm-3)"?*
**Reply**: Amended

**Reviewer 2**

**General comments**

*: In this well-written manuscript, the authors offer a new method for using MODIS cloud data combined with the CERES-CloudSat-CALIPSO-MODIS (CCCM) combined product and ERA5 reanalysis fields to assess short-term (several hour) changes in the relationships between cloud liquid water path (LWP) and cloud droplet number. The results focus on average cloud properties in 1x1 degree bins. There were a few places in the methods that need more clarification, but overall I believe the results will be helpful in understanding the factors contributing to cloud evolution over short timescales over large spatial scales, and that this information will provide an important constraint for models. I'm happy to recommend publication once the authors address the comments below.*
**Reply**: We thank the reviewer for their helpful comments and address them in turn below. Line numbers refer to the diff-ed version.

**Specific comments**

The introduction offers a nice assessment of the potential pitfalls in trying to identify aerosol impacts on the short-term development of clouds.

**Methods**: *It might be helpful to mention more explicitly that the cloud droplet number is taken only from the top layer of clouds, whereas the LWP is a column property.*

**Reply**: This is an important point, but for this case, we have restricted the study to single layer clouds, so the impact is minimised. This is now noted in L67.
* * *
*: How did the authors account for cirrus clouds? Have the authors assessed how their results might differ if they exclude pixels with multi-layer clouds (e.g., see* `https://doi.org/10.5194/amt-13-3263-2020`*)?*

**Reply**: Ice clouds are removed by selecting only gridboxes with an ice cloud fraction of less than 10%. Only single layer pixels are used when creating the $N_d$ product. As noted in the linked paper, this does not always result in the selection of single layer clouds and can produce errors in the retrieved cloud properties. We note that there is generally a good agreement between aircraft and satellite retrieved $N_d$ (Gryspeerdt et al., 2021), suggesting that these errors are small, particularly when averaged properties are used. We have included a sentence to explain these points at L67.

"Only single layer pixels and gridboxes with an ice cloud fraction above 10% are excluded to minimise the impact of thin undetected cirrus on the liquid cloud retrievals (Marchant et al., 2020)."
* * *
*: The ERA-5 wind results might really depend on altitude. Was some sort of cloud top height product used to identify the height of the cloud tops? Similarly, I am not sure how helpful surface winds are when many of the clouds of interest are at much higher altitudes (Fig. 5d).*

**Reply**: While there may be some dependence on altitude, a constant 1000hPa was used as it has been shown as effective for predicting shiptrack locations, showing it can be used for advection in the marine boundary layer. From experience in that paper, using a higher altitude frequently resulted in selecting winds above the boundary layer. This produced larger errors than using a near-surface windspeed. A sentence has been modified to explain this in more detail at L84. "at 1000hPa. This level is selected as it is able to accurately predict the locations of shiptracks given the individual ships, confirming it is suitable for calculating cloud advection over short timescales (Gryspeerdt et al, 2021)."
* * *
**Figure 1***: For clarity, it might be helpful in the figure to say something like, "Low Nd (N < 60 cm-2)" and "High Nd (Nd > 60 cm-2)" instead of just high and low Nd.*

**Reply**: We appreciate the suggestion, but feel that this might confuse things a bit. The $60 \text{cm}^{-3}$ threshold is only applied for Fig. 2, but the concepts in the Fig. 1 schematic apply to any threshold. Later parts of this work have no threshold, but still look at dLWP. A sentence has been added to the caption to make this clearer. "In sections 3.1 and 3.2, a threshold of $60 \text{cm}^{-3}$ is used to separate high and low $N_d$."
* * *
*: In the methods it said that the study is only looking at oceanic data, but terrestrial data is shown and discussed later in the paper.*

**Reply**: The focus on oceanic data is only for the CCCM data, to ensure the accuracy of the cloudsat precipitation flag. This is now explained at L93. " To

select the most accurate precipitation identification, only oceanic, liquid phase data are used over the period 2007-2011 (inclusive) are used for the CCCM part of this work."
* * *
*: It would be helpful to have an indication of sample numbers in the various figures, and to mention whether there was any cutoff for excluding locations with low sample numbers (e.g., in Figs. 4, 5, etc.), which may add uncertainty to the extreme points.*

**Reply**: Figs. 4 and 5 now include shading showing bins with fewer than 30 retrievals per bin. Also note that Fig. 4a,b have been modified so that the probabilities are normalised by the bin width (either $N_d$ or LWP).
* * *
**L.63**: *"The LWP is calculated using all the available liquid pixels, as restricting the LWP retrieval to only the pixels used for the Nd calculation biases it towards higher optical depths, leading to a high LWP bias against passive microwave LWP (Gryspeerdt et al., 2019)." That makes sense, but does not solve the problem of the Nd data then being potentially biased. Do any aircraft data support the LWP-Nd trends observed?*

**Reply**: The $N_d$ and LWP are both averaged to a 1 by 1 degree resolution, so can use different MODIS pixels to represent the $N_d$ and LWP in the same gridbox. Recent work has suggested that satellite $N_d$ retrievals are accurate in stratocumulus conditions, particularly for these averaged values (Gryspeerdt et al., 2021). We are not aware of any pure aircraft studies looking at this $N_d$-LWP relationship, although it would be an interesting area for study. However, as LWP is a column property, this would necessitate some kind of remotely-sensed retrieval, even if the $N_d$ could be measured in-situ. The following text has been added at L65 to improve clarity. "This aggregation allows different MODIS pixels to be used for the $N_d$ and LWP 1° by 1° average. Only 1° by 1° gridboxes with both a $N_d$ and LWP value are used in this work."
* * *
**Line 104**: *"This would happen even without a causal Nd impact on LWP." I got lost here. Could the authors please provide a more thorough explanation for why that might be?*

**Reply**: This effect happens due to the strong correlation between the initial $N_d$ and LWP. If this correlation is due to a retrieval bias (e.g. Gryspeerdt et al., 2019), the then initial correlation could be formed, even without a causal impact of $N_d$ on LWP. The second driver of this effect is that, on average, cases with low LWP would be expected to increase in LWP over time and those with high LWP expected to decrease LWP. We now have a situation where, without any $N_d$ impact on LWP, cases with high initial $N_d$ have a low initial LWP and hence a positive dLWP (and vice versa). This produces a correlation between $N_d$ and dLWP even though (by construction), there is no actual impact of $N_d$ on LWP or dLWP. The paragraph has been re-worded for clarity at L106 as,

"In this case, the positive $\Delta$dLWP is an artifact of the strong initial negative $N_d$-LWP relationship, as observed in previous work (Han et al., 2002; Michibata et al., 2016; Gryspeerdt et al., 2019). Over the three hour observation period, cases with a low initial LWP will tend to increase in LWP, whilst those

with a high initial LWP will decrease (a concept known as regression to the mean), returning towards an LWP steady state (Hoffmann et al., 2020). This means that on average, cases with a high initial $N_d$ will have a low initial LWP and so positive dLWP, producing the apparent positive $N_d$ impact on LWP in Fig. 1a. This relationship appears whatever the driver of the initial $N_d$-LWP relationship. If the negative relationship is produced by a feedback (e.g. wet scavenging), rather than a $N_d$ impact on LWP, it is possible to produce the observed apparent positive $N_d$ impact on LWP even if $N_d$ has no impact on LWP. By binning by the initial cloud state, this ensures that the high and low $N_d$ populations start with the same LWP, removing the impact of this regression to the mean effect"
* * *
**L117-120**: *Unless I am misunderstanding something, I don't really see enough evidence for attribution to pollution in all of the cases here. Maybe the text should be changed to something like, "is associated with pollution-dominated air masses in some locations"? For example, the Eastern Pacific area shows higher values, but is not typically thought of as a comparatively polluted region.*
**Reply**: This sentence has been amended as suggested.
* * *
**L.121**: *" clean conditions (Nd>25 cm-3 )...."I agree that low Nd levels are often associated with reduced aerosol conditions and high Nd levels are often associated with high aerosol levels, but Nd is not a direct proxy for pollution concentrations so I suggest rephrasing. Same for labelling cases with initial Nd of 25-100 cm-3 as "moderately polluted" cases.*
**Reply**: These have been modified to "low initial $N_d$" and "moderate initial $N_d$"
* * *
**Figure 5**: *Why there are different data bins in each of the 4 panels?*
**Reply**: The bins are the same size for each of the panels, but might appear different due to the varying smoothness of the fields.
* * *
**Paragraph starting on L 168**: *Sea salt is not the only source of marine CCN - in fact it may not even be the main source of marine-produced CCN depending on location and time, e.g., https://www.nature.com/articles/s41598-018-21590-9 and emissions from these other sources also may be affected by wind speed.*
**Reply**: Thank you for raising this paper, we have now added it into the explanation at L185:

" Similar relationships are likely for the emission of marine aerosol precursors (such as dimethyl sulphide, DMS), which can make up a large fraction of the CCN at low windspeeds (Sanchez et al., 2018)."
* * *
*: The authors discussed "noise." Could they please discuss in greater detail what factors contribute to this noise?*
**Reply**: This "noise" represents other processes governing the $N_d$ and LWP evolution other than the current $N_d$ and LWP state. This paragraph has been modified to start at L244

"A further complication comes from representing other factors controlling the $N_d$ and LWP evolution, other than the current $N_d$, LWP state. Random

errors in the retrievals, local variations in meteorological parameters or changes in the background aerosol properties may all have a role in the $N_d$ and LWP evolution. If these other factors are uncorrelated to the current $N_d$ and LWP"
* * *
*: Please discuss how the effects of systemic meteorological differences/changes at the different locations might affect the results. Does "noise" include this factor?*

**Reply**: Local meteorological variations that are uncorrelated to the current $N_d$, LWP are included in the "noise" factor (now included in the paragraph, see above). Larger scale variations may also impact the $N_d$ and LWP development. There are covered through the variation in the global sensitivity maps (Fig. 7), where a new joint histogram is created for each gridbox. The first sentence of section 3.6 has been modified to make this clearer (L254)

"The dLWP and d$N_d$ fields in Fig. 4 and their evolution (as in Fig. 6) vary across the globe due to variations in the background meteorological state and aerosol properties, becoming positive in some regions."

**Technical comments**
* * *
***L.62***: *"The Nd is calculated assuming an adiabatic cloud"*
**Reply**: Amended
* * *
*: Flow field is sometimes spelled flowfield. Suggest standardizing, and possibly defining at first use.*
**Reply**: Thanks, we have standardised around "flowfield".

**Bibliography**

Gryspeerdt, E., Goren, T., Sourdeval, O., Quaas, J., Mülmenstädt, J., Dipu, S., Unglaub, C., Gettelman, A., and Christensen, M.: Constraining the aerosol influence on cloud liquid water path, Atmos. Chem. Phys., 19, 5331–5347, https://doi.org/10.5194/acp-19-5331-2019, 2019.

Gryspeerdt, E., McCoy, D. T., Crosbie, E., Moore, R. H., Nott, G. J., Painemal, D., Small-Griswold, J., Sorooshian, A., Ziemba, L., Gryspeerdt, E., McCoy, D. T., Crosbie, E., Moore, R. H., Nott, G. J., Painemal, D., Small-Griswold, J., Sorooshian, A., and Ziemba, L.: The impact of sampling strategy on the cloud droplet number concentration estimated from satellite data, Atmos. Meas. Tech. Disc., https://doi.org/10.5194/amt-2021-371, 2021.

Han, Q., Rossow, W. B., Zeng, J., and Welch, R.: Three Different Behaviors of Liquid Water Path of Water Clouds in Aerosol–Cloud Interactions, J. Atmos. Sci., 59, 726–735, https://doi.org/10.1175/1520-0469(2002)059⟨0726:TDBOLW⟩2.0.CO;2, 2002.

Hoffmann, F., Glassmeier, F., Yamaguchi, T., and Feingold, G.: Liquid Water Path Steady States in Stratocumulus: Insights from Process-Level Emulation and Mixed-Layer Theory, J. Atmos. Sci., 77, 2203–2215, https://doi.org/10.1175/JAS-D-19-0241.1, 2020.

Marchant, B., Platnick, S., Meyer, K., and Wind, G.: Evaluation of the MODIS Collection 6 multilayer cloud detection algorithm through comparisons with CloudSat Cloud Profiling Radar and CALIPSO CALIOP products, Atmos. Meas. Tech., 13, 3263–3275, https://doi.org/10.5194/amt-13-3263-2020, 2020.

Michibata, T., Suzuki, K., Sato, Y., and Takemura, T.: The source of discrepancies in aerosol-cloud-precipitation interactions between GCM and A-Train retrievals, Atmos. Chem. Phys., 16, 15 413–15 424, https://doi.org/10.5194/acp-16-15413-2016, 2016.

Sanchez, K. J., Chen, C.-L., Russell, L. M., Betha, R., Liu, J., Price, D. J., Massoli, P., Ziemba, L. D., Crosbie, E. C., Moore, R. H., Müller, M., Schiller, S. A., Wisthaler, A., Lee, A. K. Y., Quinn, P. K., Bates, T. S., Porter, J., Bell, T. G., Saltzman, E. S., Vaillancourt, R. D., and Behrenfeld, M. J.: Substantial

Seasonal Contribution of Observed Biogenic Sulfate Particles to Cloud Condensation Nuclei, Sci. Rep., 8, https://doi.org/10.1038/s41598-018-21590-9, 2018.

Stephens, G. L., L'Ecuyer, T., Forbes, R., Gettlemen, A., Golaz, J.-C., Bodas-Salcedo, A., Suzuki, K., Gabriel, P., and Haynes, J.: Dreary state of precipitation in global models, J. Geophys. Res., 115, D24 211, https://doi.org/10.1029/2010JD014532, 2010.

---

## Author Response (AR2)

**Editors comment**
* * *
*: I believe that on line 60, '2011 - 2020' should be changed to '2007-2020' based on on your response to reviewer #1.*
**Reply**: Many thanks for spotting this, it has now been corrected in the manuscript.